# Validation of NASA Sea Surface Temperature Satellite Products Using Saildrone Data

**Kalliopi Koutantou [1,2,*], Philip Brunner [2]**  **and Jorge Vazquez-Cuervo [1]**

[1] NASA Jet Propulsion Laboratory, California Institute of Technology, Pasadena, CA 91109, USA
[2] Centre for Hydrogeology and Goethermics, University of Neuchâtel, 2000 Neuchâtel, Switzerland
* Correspondence: kalliopi.koutantou@unine.ch

**Abstract:** Sea Surface Temperature (SST) is at the core of many processes in the oceans. Various remote sensing platforms have been used to obtain SST products of different scales, but their validation remains a topic of ongoing research. One promising platform is an uncrewed surface vehicle called Saildrone. We use the data from eight Saildrone deployments of the USA West Coast 2019 campaign to validate MODIS level-2 and Multi-scale Ultra-high Resolution (MUR) level-4 satellite SST products at 1 km spatial resolution and to assess the robustness of the quality levels of MODIS level-2 products over the California Coast. Pixel-based SST comparisons between Saildrone and the satellite products were performed, as well as thermal gradient comparisons computed both at the pixel-base level and using kriging interpolation. The results generally showed better accuracies for the MUR products. The characterization of the MODIS quality level proved to be valid in areas covered by bad-quality MODIS pixels but less valid in areas covered by lower-quality pixels. The latter implies possible errors in the MODIS quality level characterization and MUR interpolation processes. We have demonstrated the ability of the Saildrones to accurately validate near-shore satellite SST products and provide important information for the quality assessment of satellite products.

**Keywords:** California Coast; SST; MODIS; MUR; Saildrone; quality levels; thermal gradients; remote sensing





## 1. Introduction

The sea surface temperature (SST) is a key state variable for studying and predicting the interaction between the ocean and the atmosphere [1] and an indicator of global climate change [2–4]. Its accurate mapping is critical for the understanding of ocean phenomena such as coastal upwelling. The ability to extract accurate SST and its gradients is essential for the study of coastal processes [5] and for the better understanding of the mesoscale and sub-mesoscale coastal dynamics [6]. Satellites offer the possibility to map SST at large spatial scales. Compared to in situ measurements, they provide better spatial coverage of the ocean SST [7]. However, the coarse spatiotemporal resolution of the available satellite data may not be adequate for small-scale assessments of ocean phenomena. In addition, coastal areas are often covered by clouds, resulting in data gaps within the satellite data, hence full spatial coverage is not available. For many applications the data gaps have to be interpolated, increasing the uncertainty of the final satellite product. The problem of cloud cover in coastal areas is dominant, especially in our study area, the California Coast [8,9]. In California, the term June Gloom is used to describe the cloudy, overcast sky and the cool temperatures of late spring and early summer.

Satellite SST data are provided to the users with quality flags for each pixel, typically based on a quality assessment of the atmospheric correction, atmospheric conditions at the time of acquisition, cloud cover, etc. These quality flags span a scale from 0 to 5 that indicates the quality level of each pixel. Zero indicates a complete failure or masked pixel and the highest number indicates the best quality that a pixel can have within the SST

product (https://oceancolor.gsfc.nasa.gov/atbd/sst/flag/, accessed on 1 June 2022). These per-pixel quality levels are also provided to the users within the SST product. The quality levels are very important as they are used to indicate which pixel will be used for the next processing step up to the extraction of the final SST gridded product (level-4).

Therefore, the validation of satellite-derived SST products in terms of assessing their quality levels is of critical importance. The extent to which the available satellite data can contribute to our understanding of coastal processes depends on the quality of the satellite products and the reliability of the quality levels.

Many satellites map SST at different spatial resolutions (1–25 km) using thermal infrared bands [10]. The most commonly used infrared sensors for SST are the Visible Infrared Imaging Radiometer Suite (VIIRS) instrument onboard the NOAA/NASA Suomi NPP satellite, the Moderate Resolution Imaging Spectroradiometer (MODIS) onboard the NASA Earth Observing System (EOS) Terra and Aqua satellites, the Advanced Very High-Resolution Radiometer (AVHRR) on NOAA's Polar-orbiting Operational Environmental Satellites (POES), the Along-Track Scanning Radiometer (ATSR) on the European Remote Sensing Satellite (ERS-2) and the Geostationary Operational Environmental Satellite (GOES) Imager. However, these infrared sensors are highly affected by scattering from atmospheric vapor, leading to uncertainties and data gaps in cloud-covered areas [7,11,12].

Satellite microwave sensors are also used to measure SST [13,14]. The microwave sensors record the energy emitted by the atmosphere, reflected or emitted by the ocean, or the radiance transmitted from the subsurface ($SST_{sub-skin}$), and these data are converted into temperature information. Unlike thermal infrared radiation, longer wavelength microwave radiation can penetrate through clouds, haze, and precipitation, providing SST data in all weather conditions and at night. However, the derived spatial resolution is about 25 km which may be insufficient to detect submesoscale and mesoscale variability. Some examples of microwave sensors include the Scanning Multichannel Microwave Radiometer (SMMR) carried on the Nimbus-7 and Seasat satellites, the Tropical Rainfall Measuring Mission (TRMM) Microwave Imager (TMI), the Advanced Microwave Scanning Radiometer (AMSR) instrument on the NASA's EOS Aqua satellite and the Japanese Advanced Earth Observing Satellite (ADEOS II). Microwave data have coarser spatial resolution and lower accuracy than the infrared data, and are also sensitive to surface roughness and precipitation [15].

In situ measurements to validate satellite products of SST are typically made using buoys [16–18] and Argo floats [19]. However, there are significant practical limitations [3,20]. More specifically, they only measure at a specific geographic location and therefore are unable to document mesoscale and sub-mesoscale variability. Another difference is that in situ measurements typically measure SST at a depth of a few centimeters below the surface (bulk SST/$SST_{depth}$), rather than skin or sub-skin temperature ($SST_{sub-skin}$), as is the case with thermal infrared and passive microwave sensors, respectively. The differences between $SST_{skin}$ and bulk SST increase under wind-free conditions or when large amounts of incoming solar radiation reach the ocean surface. As a result, there may be differences between skin temperature and temperatures at several meters of depth. These differences should be taken into account when validating satellite measurements derived from either thermal infrared or passive microwave sensors.

A recently developed uncrewed surface vehicle (USV) technology called Saildrone samples the ocean surface every minute and provides multiple measurements of the ocean surface, including the skin ($SST_{skin}$) and the SST a few centimeters below the surface (bulk SST/$SST_{depth}$) (https://www.saildrone.com/, accessed on 1 June 2022). The ability of the Saildrone to sample the ocean surface with high temporal resolution and increased spatial coverage due to its autonomous, rapid navigation allows for a better comparison with satellite data than the traditional in situ measurements mentioned above. Saildrones also allow the extraction of spatial gradients of the ocean parameter of interest due to their high spatiotemporal resolution [5].

To observe dynamic coastal phenomena such as the upwelling at small scales, a combination of infrared, microwave, and in situ data is promising. In this context, NASA's

Multi-scale Ultra-high Resolution (MUR) SST product of the Group for High-Resolution Sea Surface Temperature (GHRSST) combines data from different sources. The current version of MUR (Version 4.1, http://dx.doi.org/10.5067/GHGMR-4FJ04, accessed on 1 June 2022) combines MODIS SST retrievals at a high spatial resolution of about 1 km, Advanced Very High-Resolution Radiometer (AVHRR) infrared SST retrievals at a medium resolution of 4 to 8.8 km, and microwave SST retrievals at a coarser spatial resolution of 25 km. This combination aims to fill the data gaps in areas where only infrared or microwave data are available [21]. The approach used in the MUR data maximizes the use of the infrared data, where available, from MODIS (level-2 product) and AVHRR. MUR is globally gridded at 1 km resolution and available in daily maps using an interpolation technique based on a wavelet decomposition [22].

This paper presents a validation of MODIS level-2 (L2P) and MUR level-4 SST products over the California Coast. We also examine the MODIS level-2 quality levels characterization along the California Coastal region. This is the first study to compare MODIS level-2 (L2P) with MUR level-4 SST products over the California Coast. It is also the first study that attempts to validate MODIS level-2 (L2P) data from a spatiotemporal perspective by applying the different quality levels off the California Coast. MODIS L2P data are the main input for the production of the level-4 gridded SST MUR products and therefore the validation of their quality is crucial. Their quality is documented by the quality levels that have been established during the primary data processing. MUR uses only the MODIS pixels of good-quality (quality level 5). Otherwise, the interpolation to fill the pixels where no MODIS data were found, or where quality level 5 pixels are not available, is done using data of coarser spatial resolution. More specifically, it uses AVHRR data from 4 km to 8.8 km or microwave data from the Advanced Microwave Scanning Radiometer (AMSR-E), the WindSat Polarimetric Radiometer, and the Advanced Microwave Scanning Radiometer 2 (AMSR2), as well as in situ data [22] are used.

Assessing the quality levels of MODIS level-2 data is important because incorrectly characterizing a pixel as "bad-quality" or "good-quality" will lead to incorrect exclusion or inclusion, respectively, of that particular pixel from the final SST product. In the former case, erroneous exclusion can lead to unneeded data interpolation thus reducing the accuracy of the final SST product. The validation of both MODIS level-2 and MODIS level-4 as well as of the quality levels of MODIS level-2 was carried out using the Saildrone data due to its ability to sample the sea surface of coastal areas at high spatial and temporal resolution. The validation was performed at the pixel level. Gradients were also calculated to test the ability of 1km satellite products to correctly estimate thermal gradients. The gradients are of particular importance because of they assist the understanding of the coastal SST patterns and the associated upwelling phenomena. Despite the rapid coverage of large areas that Saildrones provide, they are often only launched close to the coast. Therefore, the ability to interpolate Saildrone-based SST values in unsampled areas outside the coast can help to validate satellite products over a larger spatial extent. To achieve this, this study used kriging interpolation to predict the Saildrone SST values in unsampled locations by taking into account the spatial correlation between sampled SST values [23].

The choice of the California Coast as the study area was made because it is a highly productive coastal ecosystem due to the California Current [24,25]. Coastal upwelling is the dominant physical phenomenon affecting production in the California Current system. It occurs from April to September when strong northerly winds blow along the Oregon and Washington coasts. This seasonal cold water is associated with high production by phytoplankton which is the base of the food chain, attracting large numbers of consumers and leading to one of the most productive fisheries in the world. These fisheries play a key role in the economy and culture of the California Current (https://ecowatch.noaa.gov/regions/california-current).

In this current study, we focus specifically on:

- Per-pixel validation of SST values from satellite products using Saildrone data
- Assessing the quality levels of MODIS level-2 data

- Estimation of the SST gradients using pixel-based and kriging approaches from the Saildrone data and comparison with those estimated directly from the satellite products.

In the following sections, we present the methods we used for the SST and gradient validations, as well as a description of the results and discussion compared to previous studies.

## 2. Materials and Methods

### 2.1. Study Area

The study area covers part of the North Pacific Ocean along the California Coast, from 46.783603, −126.339794 north to 24.178057, −116.711086 south, in geographic coordinates (see Figure 1a). The area includes the coastline from Washington to Baja California, where the dominant coastal upwelling occurs.

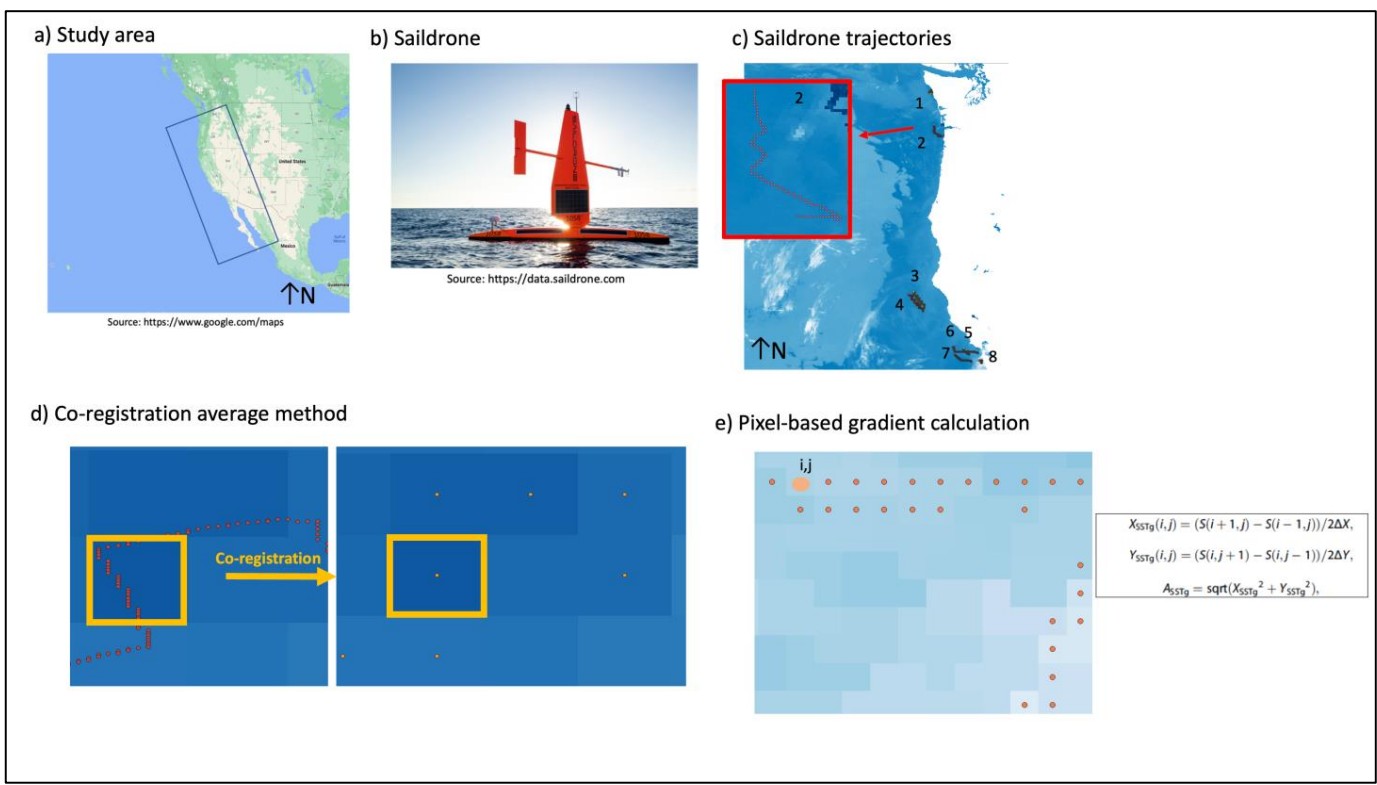

**Figure 1.** Study area; (**a**) Saildrone vehicle; (**b**) The eight West Coast 2019 Saildrone trajectories along the study area; (**c**) Co-registration average method for the pixel-based SST calculation (**d**) Formula for pixel-based gradient calculation along each of the pixels within the Saildrone trajectory (**e**).

### 2.2. Data

#### 2.2.1. In Situ Measurements

In this study, as a reference for the validation of the satellite products we used the data from the Saildrone USA West Coast 2019 campaign, which integrates eight deployments along the California Coast (see Figure 1c). The USA West Coast 2019 deployment covers spatially the area from 31.78 to 48.5 degrees latitude and −127.62 to −118.36 degrees longitude. Data collection took place from mid-June to mid-September (18 June 2019 to 16 September 2019). The eight Saildrone deployments followed different trajectories between each other, continuously sampling the ocean surface every minute for the entire period between 18 June 2019 and 16 September 2019. We selected only the Saildrone measurements for each deployment corresponding to the dates of 27 and 30 of June, 7th of July, and 11th of August (from 0:00 to 23:59 p.m. for each date), as these are the dates of the satellite data used for this study (see next section).

We used the bulk SST/SST$_{depth}$ measurements from the CTD sensor at 0.6 m below the sea surface level (TEMP_CTD_RBR_MEAN value of SST) because at the time of the current study, the Saildrone sensor measuring bulk SST/SST$_{depth}$ has proven to be more accurate than the one measuring SSTskin. The instrumentation on Saildrones is not a new technology but is based on the application of standard Conductivity/Temperature/Depth measurements. These instruments have been used extensively in ships, buoys, etc. The infrared radiometers on Saildrones were not used in the study, because there are still calibration issues. Thus, for this study, only the CTD sensors were used. CTD sensors are used on the global ARGO array as a reference for many data sets, including satellite-derived sea surface temperature and salinity [26].

2.2.2. Satellite Sea Surface Temperature Products

We used two different SST satellite products at 1 km spatial resolution over the California Coast for the dates of the 27th and 30th of June, the 7th of July, and the 11th of August 2019, which coincide with the Saildrone data. More specifically, the two datasets used are as follows:

The first dataset is the GHRSST level-2P Global Skin Sea Surface Temperature from the Moderate Resolution Imaging Spectroradiometer (MODIS) on the NASA Aqua satellite. This skin SST product is produced from both day and night observations, derived from the long-wave IR 11- and 12-micron wavelength channels of Aqua MODIS (https://podaac.jpl.nasa.gov/dataset/MODIS_A-JPL-L2P-v2019.0, accessed on 1 June 2022). In addition to the skin SST data, we used the quality level information that accompanies the MODIS level-2P products, which includes information on how good/trustworthy the SST value of each pixel is. Higher values correspond to better-quality pixels. This information is very useful because bad-quality pixels (e.g., cloud-covered pixels) are masked out and only the good-quality pixels (quality level 5) are used during the interpolation process to produce the final gridded MUR product (level-4 SST map).

The second dataset is the GHRSST level-4 MUR Global foundation SST product, which combines nighttime GHRSST L2P skin and sub-skin SST observations from several instruments, including the NASA Advanced Microwave Scanning Radiometer-EOS (AMSR-E), the JAXA Advanced Microwave Scanning Radiometer 2 on GCOM-W1, the MODIS on the NASA Aqua and Terra platforms, the US Navy WindSat microwave radiometer, the AVHRR on several NOAA satellites, and in situ SST observations from the NOAA iQuam project [20].The iQuam data are used for bias correction. The level-4 product is a daily global map gridded at 0.01° spatial resolution (https://podaac.jpl.nasa.gov/dataset/MUR-JPL-L4-GLOB-v4.1, accessed on 1 June 2022).

*2.3. Sea Surface Temperature Comparisons*

To validate the satellite products using the Saildrone data, we performed a pixel-based comparison of the SST values of the three datasets, as well as a validation of the MODIS level-2 quality levels for each MODIS pixel.

To perform these two tasks, the three datasets had to be georeferenced to each other to eliminate spatial discrepancies between the datasets. As the MODIS level-2 product is not georeferenced (coordinates are on the native satellite swath grid), we registered it to the MUR level-4 product within the SeaDAS software provided by NASA (https://seadas.gsfc.nasa.gov/, accessed on 1 June 2022). Due to the high temporal sampling resolution of the Saildrone (1 minute), many Saildrone points fell within the same MODIS and MUR pixel.

Therefore, a further step was to co-register the Saildrone points with the satellite products. The co-registration was performed using the average of all Saildrone SST point values falling into each MODIS and MUR pixel [5] (see Figure 1d). After the co-registration of the three datasets, we performed the pixel-based comparisons of the SST values, and their discrepancies were quantified in terms of root mean squared error (RMSE) and mean absolute error (MAE).

*2.4. Thermal Gradients Comparisons*

Two different workflows were used to calculate and compare the gradients between the three datasets:

### 2.4.1. Pixel-Based Approach

The first approach is based on a pixel-based comparison between the gradients calculated on the Saildrone pixels (pixels along each Saildrone trajectory) and the gradients calculated on the corresponding pixels in the satellite products (see Figure 1e).

To perform this comparison, we calculated the magnitude of the SST spatial gradient at each pixel location i, j for all three datasets using finite central differences [5]. The formula is as follows:

$$X_{SSTg}(i,j) = (S(i+1, j) - S(i-1, j))/2\Delta X$$
$$Y_{SSTg}(i,j) = (S(i, j+1) - S(i, j-1))/2\Delta Y \tag{1}$$
$$A_{SSTg} = \sqrt{X_{SSTg}(i,j)^2 + Y_{SSTg}(i,j)^2}$$

The terms $X_{SSTg}$ and $Y_{SSTg}(i,j)$ denote the corresponding gradient (in K/km) in the longitudinal and latitudinal directions, respectively, and $S(i,j)$ denotes the SST pixel value of each product after its co-registration (MODIS, MUR, Saildrone) at location i (longitude), j (latitude) along the Saildrone trajectory. The terms $\Delta X$ and $\Delta Y$ are the distances in km the longitudinal and latitudinal directions between neighboring pixels. $A_{SSTg}$ is the magnitude of the SST gradient at the given pixel at the given i, j location.

The discrepancies between the pixel-based calculated gradients for each of the satellite products and the Saildrone were quantified in terms of RMSE and MAE.

### 2.4.2. Kriging-Based Approach

The second approach was based on a kriging interpolation workflow for predicting Saildrone SST in unsampled locations [23]. This interpolation workflow is used to predict the SST in a larger spatial area around the sampled pixels where in-situ data were not available. This allows us to generate in situ data in unsampled areas and thus validate the satellite products in larger spatial areas. For the interpolation, we chose the ordinary kriging method. Kriging interpolation allows the spatial correlation between the sampled values to be taken into account and is therefore particularly useful in the case of our application.

More precisely, spatial correlation simply refers to the fact that nearby pixels are expected to have more similar SST values than those further away. Therefore, the interpolation is based on the spatial arrangement of the empirical observations (in our case the SST sampled Saildrone values), rather than using an assumed model of spatial distribution.

The SST Saildrone data were used to derive the experimental variogram. The full workflow for the kriging interpolation is shown in Figure 2. The experimental variogram is a discrete function calculated using a measure of the variability between pairs of points at different distances. In other words, it describes how the data are correlated with distance.

As the experimental variogram does not provide information for all possible directions and distances between points, we fitted a variogram model to approximate the experimental variogram. In a subsequent step, the kriging interpolation was performed to predict the SST values at the unsampled locations using the fitted variogram. The prediction was made in the spatial area/window around the pixels of each Saildrone trajectory, as described in the previous paragraph.

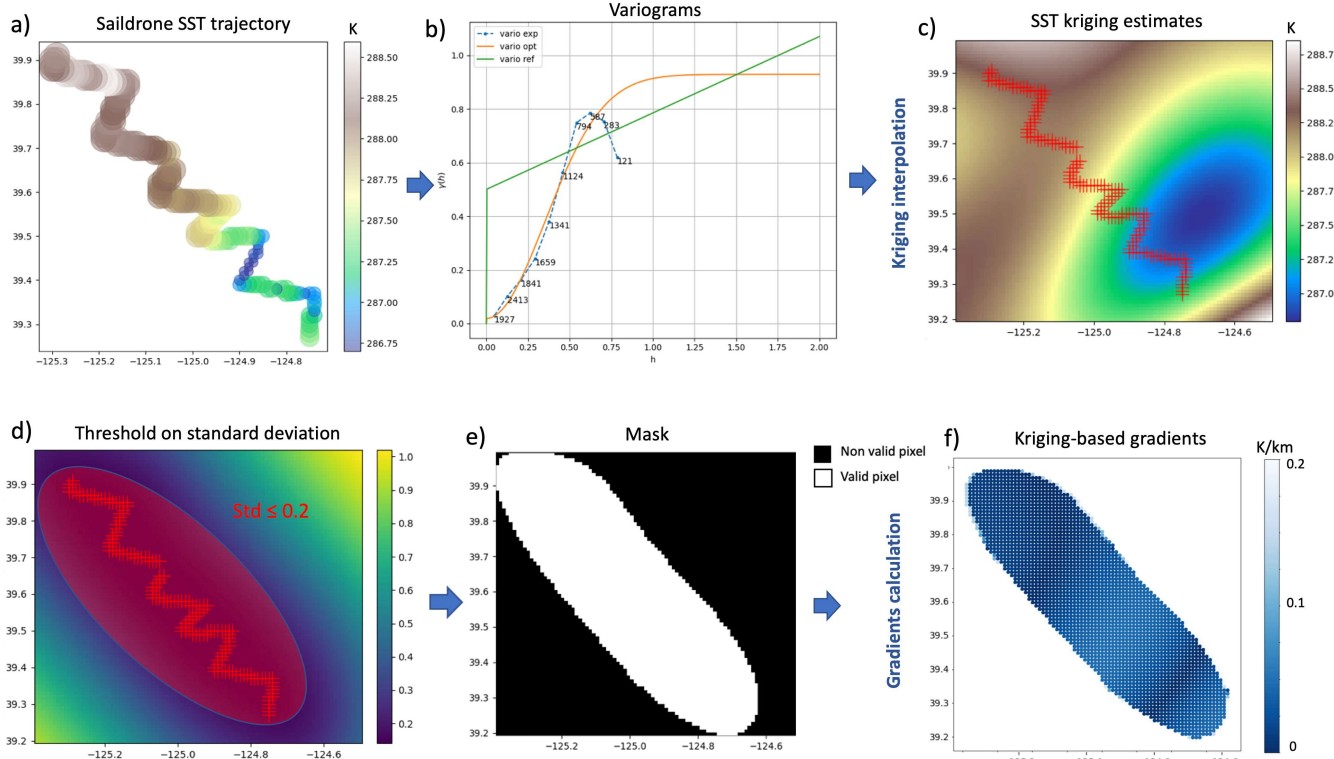

**Figure 2.** Workflow for the kriging interpolation using the Saildrone SST values for one of the trajectories.

The standard deviation of the interpolated SST values was also estimated at each predicted pixel location. Based on the standard deviation and considering that SST estimations closer to the locations of the sampled data are more accurate, a mask was created. The mask included pixels with a standard deviation less than a defined threshold (see Figure 2d,e). This threshold was chosen to retain only the pixels with the smallest standard deviation, i.e., pixels close to the sampled locations/pixels. Gradients were then calculated for the valid pixels within the mask using Formula (1).

Using the same mask, gradients were calculated for the same pixels on each satellite product using Formula (1) and their differences between those calculated on the kriging-based predicted SST pixel values as described above were expressed in terms of RMSE.

## 3. Results

### 3.1. Sea Surface Temperature

The comparisons between Saildrone and the two satellite products are summarized in the following two sections: In the first one, the accuracies in terms of RMSE and MAE are analyzed for all the Saildrone trajectories and are compared between the two products. The analysis is done with respect to the quality level of the corresponding MODIS product. The second section shows the SST maps for all the dates for both products, as well as the quality level of each pixel of the corresponding MODIS product. In addition, in this second section we show the evolution of the SST for both products along two characteristic trajectories, as well as the per-pixel quality level value of each MODIS pixel along these two trajectories. The second section aims to understand the differences between the three datasets in space.

Each of the Saildrone trajectories was classified based on the number of bad-quality MODIS pixels it contained. As mentioned in the previous section, MODIS pixels of quality level one are defined as "bad-quality pixels". Pixels of quality levels three and four are defined as "medium-quality pixels" and pixels of quality level five are mentioned as "good-quality pixels". The percentage of bad-quality MODIS pixels for each trajectory is relative to all the pixels along each trajectory.

In the following, trajectories containing only bad-quality MODIS pixels (100%) are defined as "bad-quality trajectories". Trajectories containing more than 30% and less than 100% bad-quality MODIS pixels are defined as "medium-quality trajectories", and trajectories containing equal or less than 30% bad-quality MODIS pixels are defined as "good-quality trajectories". These classification thresholds are derived from the data to have a very well-defined separation between the three classes. The trajectories classified above are colored in red, orange, and green, respectively, in the graphs and maps of the following sections.

### 3.1.1. Pixel-Based Sea Surface Temperature Comparisons

Figure 3a,b show the RMSE and MAE between the Saildrone and the two satellite products for all 23 Saildrone trajectories. Figure 3c shows the percentage of bad-quality pixels for each of the trajectories. Bad, medium, and good-quality trajectories are shown in red, orange, and green colors respectively, according to the classification described at the beginning of Section 3.1. From all the quality trajectories, we extracted a mean RMSE of 0.86 and 4.64 K (Kelvin; hereafter K stands for Kelvin) for MUR and MODIS, respectively.

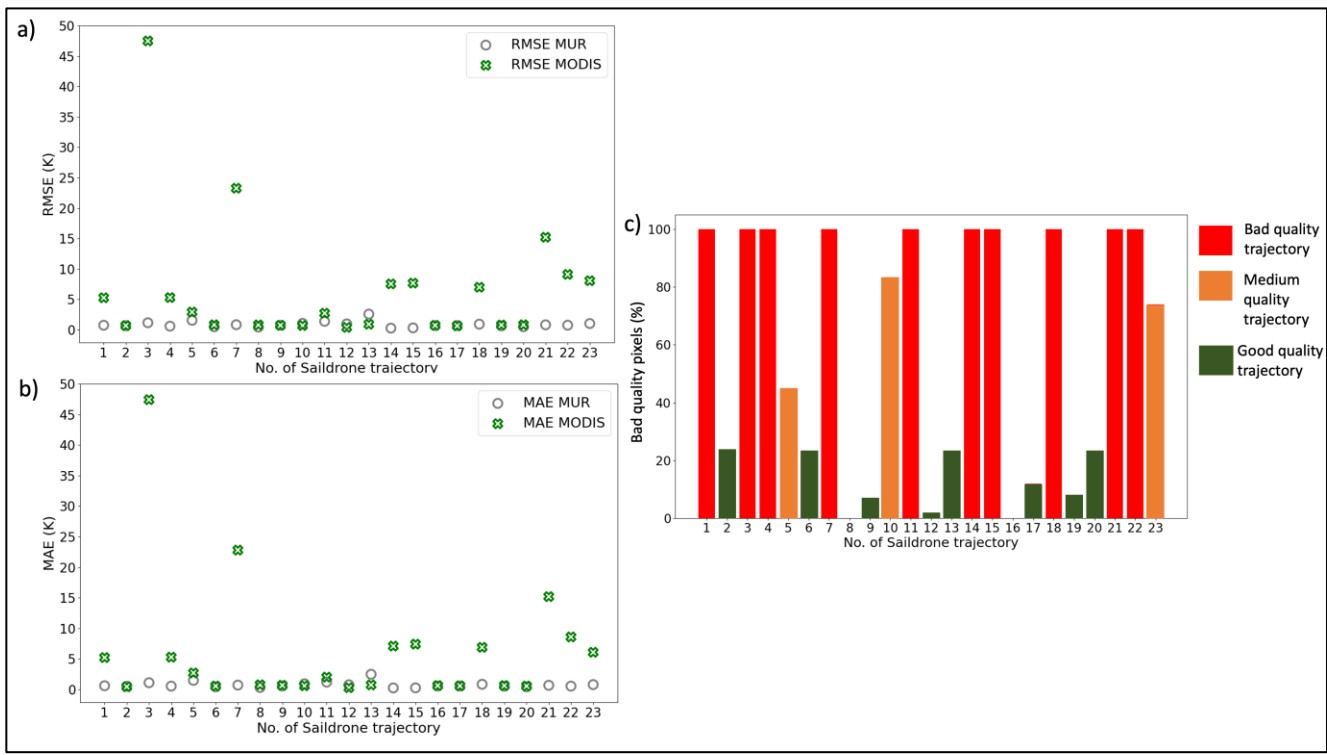

**Figure 3.** RMSE (**a**) and MAE (**b**) plot for both products and all the trajectories; percentage of bad-quality MODIS pixels for all the trajectories: in red, orange, and green the bad, medium, and good-quality trajectories, respectively (**c**).

For all of the bad-quality trajectories, Figure 3a,b show that the MODIS RMSE and MAE are higher than the corresponding RMSE and MAE for the MUR. In particular, in some cases, the MODIS RMSE and MAE reach very high values (trajectories with numbers 3, 7), even almost 50 K. This result indicates the very poor quality of the MODIS level-2 product, especially in areas close to the coast. However, since the other trajectories have much lower RMSE and MAE values than the 50 K value, we can consider this value as an outlier.

For the medium-quality trajectories, the RMSE and MAE values for the two products are more similar. Thus, as the quality of the MODIS product increases, its accuracy does not differ much from the accuracy of the corresponding level-4 interpolated MUR product. In particular, for one of the medium-quality trajectories (number 10), the accuracy of the MODIS product is better than that of the MUR.

However, for the medium-quality trajectories (numbers 5, 10, 23), the MUR products have higher inaccuracies than other bad-quality trajectories, even though they contain better-quality MODIS pixels. The same is true for the MODIS product for trajectory number 23. This specific result implies either the existence of errors during the interpolation process to extract the level-4 MUR product or the misclassification of MODIS pixels with respect to their quality level. In the latter case, bad-quality MODIS pixels may have been misclassified as medium-quality pixels and therefore the medium-quality trajectories contain fewer bad-quality pixels than they actually do.

For the good-quality trajectories, the RMSE and MAE values are generally lower for both products and more similar for both products, as can be seen in Figure 3a,b. Especially for the MODIS products, the accuracies are much better than the accuracies of the bad-quality trajectories

Therefore, as the quality of the MODIS products increases, MUR and MODIS have more similar RMSE and MAE values. There are cases of two trajectories (numbers 12, 13) where the MODIS is more accurate than the MUR. However, as in the case of the medium-quality trajectories, some good-quality trajectories appear to have higher RMSE and MAE values for the MUR products than other medium and bad-quality trajectories (such as numbers 12, 13). In addition, for the MODIS products, some good-quality trajectories (such as numbers 6, 8, and 13) are less accurate than the medium-quality trajectory number 10. These last results confirm the similar finding derived for the medium-quality trajectories described in the previous paragraph regarding the potential errors within the processes of MODIS quality level characterization and MUR level-4 interpolation (see more in Section 4).

### 3.1.2. Sea Surface Temperature Maps

Figure 4 shows the MODIS and MUR SST maps and the Saildrone SST trajectories for all the trajectories and for all the dates. It also includes the quality level map for the same area, where bad-quality MODIS pixels are in red, medium-quality MODIS pixels are in orange, and good-quality MODIS pixels are in green. The numbering of the trajectories follows the numbering of the trajectories in the plots of the previous section.

Comparing the MODIS and MUR maps and using the quality level maps for all the dates, we see that the differences between the SST values within the products are greater in areas of bad-quality pixels (red areas). These areas are mainly near the coast (land is shown in black). Interestingly, the MODIS SST values appear to be underestimated in these red areas, as they are smaller than the corresponding MUR values (white areas within the MODIS maps). Therefore, the MODIS products seem to be less accurate in the red areas. The differences between the two products become smaller (more similar pixel colors) in the medium and good pixel areas, shown in orange and green colors, respectively. Especially in the green areas, the two products seem to have the smallest differences in SST, even though they are spatially limited. Consequently, the visual interpretation of the maps in Figure 4 confirms the results analyzed above on the basis of Figure 3.

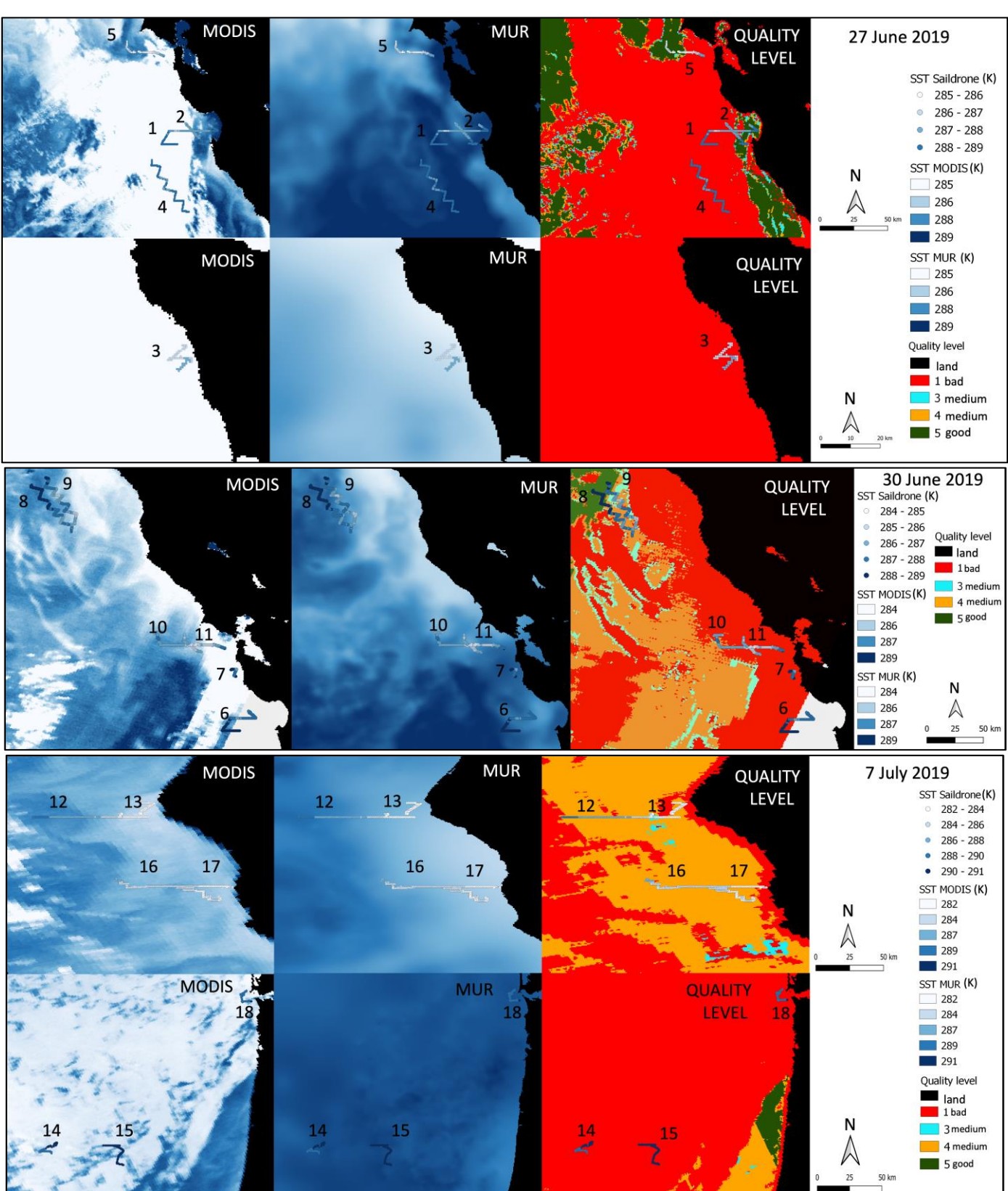

**Figure 4.** *Cont.*

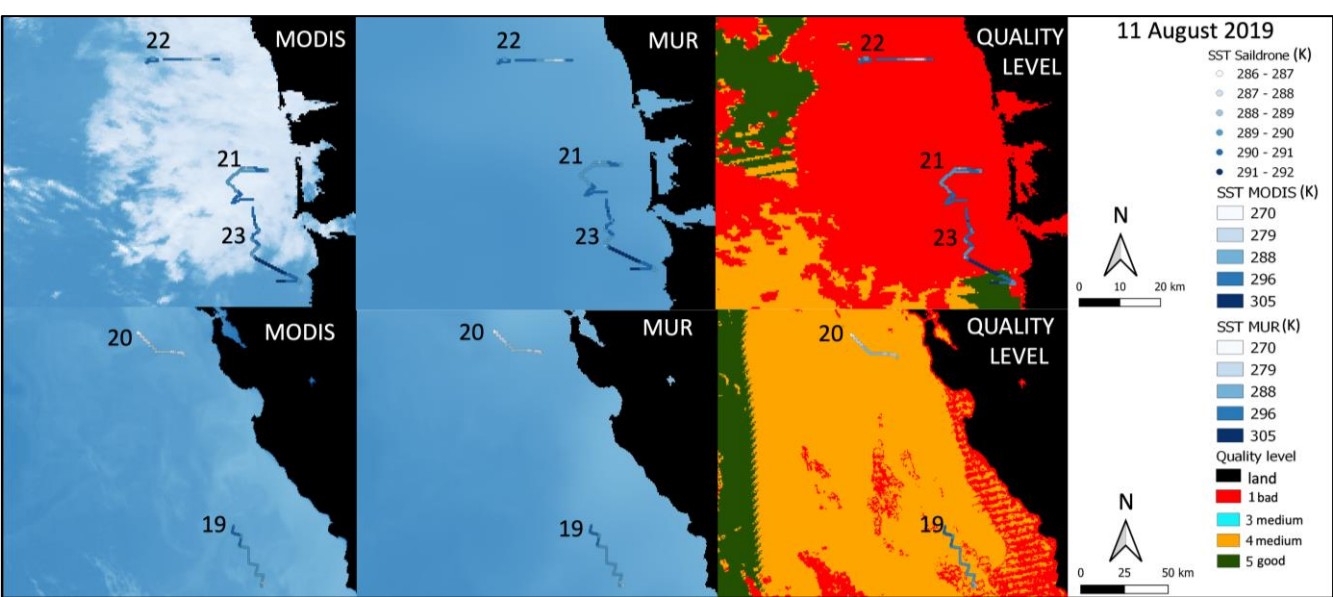

**Figure 4.** SST and quality level maps for all the dates and for the three datasets. The SST maps depict the SST values of the two satellite products and the Saildrone trajectories in K. The quality level maps correspond to the quality level (from 1 to 5) for each MODIS pixel.

### 3.1.3. Sea Surface Temperature Evolution along Trajectories

The plots in Figure 5 show the evolution of the SST for the three datasets along two Saildrone trajectories during the 11th of August (plots 1a, 2a). The quality level of each MODIS pixel along each of the trajectories is also shown in plots 1b and 2b. The trajectories are depicted on a map that shows the MODIS quality levels of each pixel in the surrounding area (plots 1c, 2c). The first trajectory (plot 1) shows a medium-quality trajectory, as it contains 74% bad-quality pixels. The second trajectory (plot 2) shows a good-quality trajectory, containing only 8% bad-quality pixels.

Firstly, the MUR SST values (green color in plots 1a, 1b) are closer to the Saildrone reference SST values (in blue) than to the MODIS values (in orange) over most of the trajectory and for both trajectories. Secondly, the MODIS signal is noisier than the MUR throughout the entire trajectory for both trajectories.

Plots 1b and 2b show for both trajectories that the mismatch between MODIS and Saildrone increases for pixel locations that correspond to bad-quality level MODIS pixels (quality level 1). These areas are marked with red boxes in plots 1b, and 2b.

For the medium-quality trajectory, the mismatch between Saildrone and MODIS increases at the end of the trajectory where only bad pixels exist, and reaches about 20 K. For the good-quality trajectory, the largest mismatch is found at the beginning of the trajectory (around 3 K), where the bad pixels are mainly located. For the medium-quality trajectory, the mismatch between Saildrone and both products becomes smaller at the locations of good-quality level pixels which are shown in green color in map 1c. These good-quality areas are located at the beginning of the trajectory. In the case of the good-quality trajectory, there are mismatches between both products and Saildrone along the entire trajectory. However, the mismatches decrease at the locations of the medium-quality level pixels, as shown in orange in map 2c.

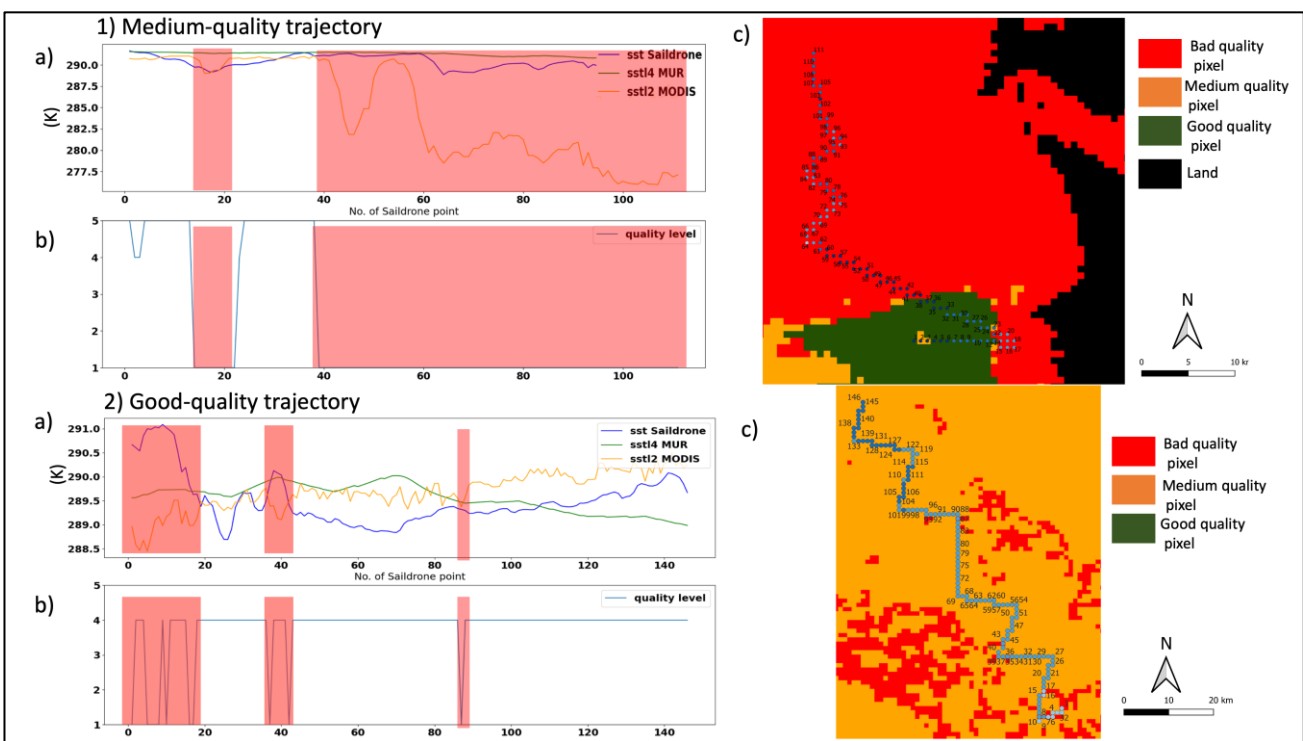

**Figure 5.** SST evolution of the Saildrone (blue), MUR (green), and MODIS (orange) along two Saildrone trajectories during the 11th of August (**1a**,**2a**); the quality level of each MODIS pixel along the trajectories (**1b**,**2b**); the quality level maps (**1c**,**2c**) with the trajectories, in which the points of the trajectories are numbered following the numbering of the X-axis in the plots (**1a**,**2a**) and (**1b**,**2b**).

### 3.2. Thermal Gradients

### 3.2.1. Pixel-Based Approach

For the pixel-based approach, Formula (1) was used to calculate the spatial SST gradient for each of the pixels along the Saildrone trajectories and the corresponding pixels within the two satellite products. Figure 6a summarizes the results in terms of the RMSE between Saildrone and the two satellite products, relative to the percentage of bad-quality MODIS pixels of each trajectory shown in Figure 6c. From all the quality trajectories we extracted a mean RMSE of 0.19 and 0.84 K/km for MUR and MODIS, respectively.

For all the bad-quality trajectories, the MUR products are more accurate in terms of RMSE than the corresponding MODIS products, except for one trajectory (number 12). The RMSE values range from 0.16 to 4.6 K/km for MODIS and from 0.04 to 0.39 K/km for MUR. The differences in RMSE between the two products are large, reaching almost 4.2 K/km. The MODIS products appear to have high inaccuracies (RMSE up to 4.6 K/km) in some bad-quality trajectories (numbers 3, 7).

For the medium and good-quality trajectories, the RMSE values for both products are closer to each other than in the case of the bad-quality trajectories. MUR is more accurate than MODIS. However, the trajectories of medium-quality do not correspond to lower RMSE values than the bad-quality trajectories for both products. On the contrary, for some trajectories (numbers 5, 23) the inaccuracies are higher compared to some bad-quality trajectories, for both products and especially for MUR.

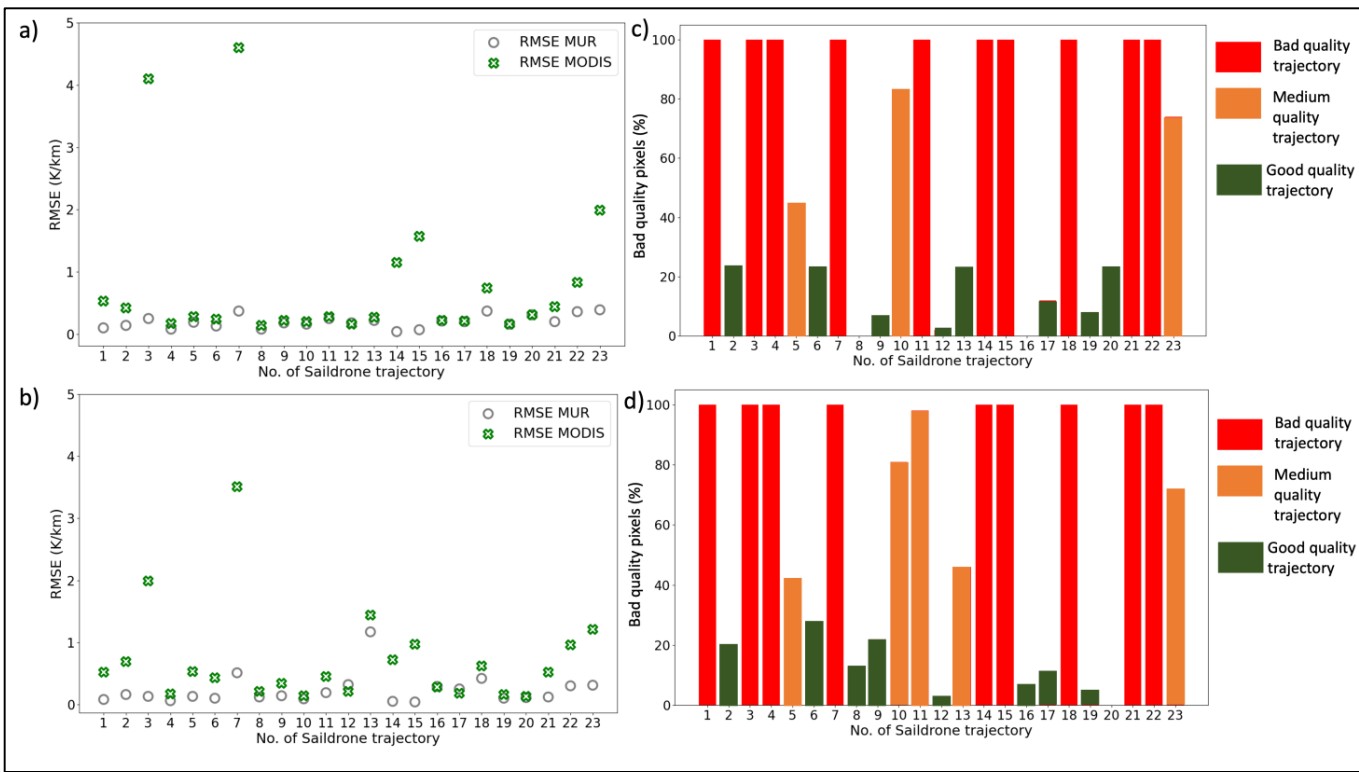

**Figure 6.** RMSE plots for the pixel-based (**a**) and kriging-based (**b**) gradients comparisons; percentage of bad-quality MODIS pixels for each of the trajectories for the pixel-based (**c**) and kriging-based (**d**) approach: in red, orange, and green the bad, medium and good-quality trajectories, respectively.

Regarding the good-quality trajectories, the accuracies for MODIS for the good-quality trajectories are generally much better than those for the bad-quality trajectories. The only exception is trajectory number 2, where the RMSE is larger than the RMSE of one of the medium-quality trajectories (number 5) and one of the bad-quality trajectories (number 11).

Comparing the two products, MUR is generally more robust or equally robust compared to MODIS, except for trajectory number 12.

The higher RMSE for the MUR compared to the MODIS product for the good trajectory number 12, and the increased RMSE values for some medium and good-quality trajectories for both products, are similar results to those derived from the SST pixel-based comparisons in Section 3.1.1. They may indicate the following: Either possible errors in the characterization of the quality levels of some MODIS pixels, or errors in the interpolation to produce the level-4 MUR product.

### 3.2.2. Kriging-Based Approach

For the kriging-based approach, Formula (1) was used to calculate the spatial SST gradients for all the kriging-based SST values within the mask, as described in Section 2.4.2. Formula (1) was also used to calculate the gradients for all the pixels of the two satellite products within the same mask. The comparisons in terms of RMSE between the kriging-based gradients and those calculated directly from the satellite products are shown in Figure 6b. The percentage of bad-quality MODIS pixels of each trajectory is also shown in Figure 6d. From all the quality trajectories we extracted a mean RMSE of 0.23 and 0.71 K/km for MUR and MODIS, respectively.

The results in terms of RMSE are slightly better for the kriging-based approach than for the pixel-based approach for MODIS products, and very similar for MUR products (slightly better for the pixel-based approach). The RMSE values range from 0.13 to 3.51 K/km for MODIS and from 0.04 to 1.17 K/km for MUR. The maximum difference between the two products is 3 K/km.



For all the bad-quality trajectories the MUR RMSE is lower than the RMSE of the MODIS. The highest inaccuracies of MODIS are found mainly for the bad trajectories, with a maximum RMSE of 3.51 K/km (number 7). The differences between the two products decrease for the medium and good-quality trajectories. For the medium-quality trajectories, MUR is generally more accurate than MODIS. For the good-quality trajectories, MUR is generally more accurate than the corresponding MODIS, except for the trajectories with numbers 12, 16, and 17. Additionally, as found in the results for the gradient pixel-based approach, some medium and good trajectories (numbers 2, 11, 23) have higher MUR errors than some bad trajectories.

These latter results, also derived based on the pixel-based SST comparisons and the pixel-based gradient comparisons, suggest possible biases in the processes of characterizing the quality levels and the MUR interpolation.

## 4. Discussion

### 4.1. Accuracies Compared to Previous Saildrone Studies

We derived mean SST RMSE of 0.86 and 4.64 K for MUR and MODIS, respectively (excluding the outlier of 50 K).

Our study showed higher bias and RMSE between SST MUR and Saildrone for most of the trajectories compared to a previous study conducted along the Baja California (bias of 0.32 deg Celsius and RMSE of 0.42 deg Celsius) [27]. However, compared to another study that compared MUR SST with two Saildrone deployments over the Arctic and Alaskan coastal waters which derived RMSE of about 7 and 9 deg Celsius [28], our results are more promising in terms of the accuracy of the MUR products (we found a maximum RMSE of 2.57 K).

The high gradient values we obtained in most trajectories are due to the coastal upwelling that occurs within the distance of 100 km from the California Coast. An earlier study over the California Coast derived SST gradients greater than 0.1 deg Celcius/km [29]. Similar results for the same area are derived from the recent study by Vazquez-Cuervo et al., 2023 who found gradients for MUR products as high as 1 K/km [5]. They also found very similar results for the MUR RMSE (0.21 K/km) compared to ours (mean RMSE MUR of 0.19 K/km for the pixel-based gradient approach and 0.23 K/km for the kriging-based approach). Therefore, our RMSE values for the MUR gradients are reasonable considering the ocean dynamics of the California Coast due to the California Current System. Comparing the gradients between MUR and Saildrone over Peru and Chile, ref. [30] identified the maximum magnitude of the gradients up to 0.03 K/km. In our study, the maximum magnitude of the MUR gradient was higher, even reaching 0.8 K/km.

Since no previous studies have validated MODIS level-2 over the California Coast, we were not able to compare our results with previous studies specifically for our study area. The mean SST MODIS RMSE of 4.64 K and the mean gradient RMSE of 0.71–0.81 K/km are rather high compared to the MUR RMSE of our study and the previous studies over the California Coast. These high inaccuracies indicate the poor quality of MODIS level-2 data in coastal areas, especially over summer months when there is increased cloud cover. In the next section, we compare our results for MODIS with previous MODIS studies conducted in other areas around the world.

### 4.2. Accuracies Compared to Previous MODIS Studies

Comparing our MODIS results in terms of RMSE SST with previous studies conducted in other study areas, we conclude that our MODIS RMSE values are rather high.

More specifically, ref. [31] found a bias of 0.19–0.34 deg Celsius within the MODIS data in the South China Sea when compared with in situ data 5 m below the surface. Comparison with coastal buoys extracted RMSE between 0.36 and 1.42 deg Celsius over the Bay of Bengal, India [32], lower than in our study. Ref. [33] compared MODIS level-2 with $SST_{depth}$ buoys-based measurements over the Persian Gulf and found a bias of 0.07 deg Celsius at 63 measurement points and a RMSE of 0.53 deg Celsius.

The high RMSE and MAE values of the MODIS products that were found close to the coast, confirm the poor quality of the infrared data in cloud-covered areas, as also found in the study by [18]. In fact, in the vicinity of the coast, there are only bad-quality pixels due to high cloud cover during the summer months. We have also seen in Section 3.1.3 that the MODIS signal is very noisy and noisier than the corresponding MUR signal. One reason that could explain the high noise within the MODIS level-2 product, is the fact that since it is a level-2 product it is still at an early stage of processing where not all noise sources have been eliminated. Other reasons that may explain the much noisier MODIS signal, such as errors related to sensor characteristics are subjects for future investigations and are beyond the scope of this study.

Regarding the MUR products, beyond the comparisons we made between them and the Saildrone and MODIS data, we could not perform further analysis to assess the per-pixel quality of the interpolation as we did for the MODIS quality levels. This is because we did not have information available on exactly which data (AVHRR, microwave, in situ) were used to interpolate each pixel when MODIS level-5 data were not available.

### 4.3. Source of Errors Due to Saildrone

Existing literature has already demonstrated the efficiency of averaging Saildrone values for co-registration with the satellite products [27–29]. The Euclidean distance co-registration method was also tested on our dataset and gave similar results to the averaging co-registration method. The Saildrone is a wind-powered vehicle and therefore shifts in its trajectory are expected. Although no SST differences of more than half a K were found within pixels for the Saildrone trajectories, averaging for co-registration purposes could also introduce uncertainties. In addition, the Saildrone can detect SST changes at higher spatiotemporal resolutions (sub-kilometer and sub-daily) than the corresponding satellite products, which also explains the discrepancies [27].

### 4.4. Source of Errors Due to Mismatches between the Instruments

There are differences in what the three instruments explicitly measure, both in space and in time. MODIS L2P maps $SST_{skin}$ whereas MUR is tuned to a bulk temperature by using only nighttime retrievals. Thus, the issue of diurnal variability could still be a factor in the measured differences between the three data sets. Saildrone sensor used in this current study measures bulk $SST/SST_{depth}$ at 0.6 m below the surface. The Saildrone sensor that measures $SST_{skin}$ is also under development, according to a recent study [34], and this will also improve the ability of Saildrones to validate satellite SST products.

In addition, MUR uses a 5-day window for the interpolation, unlike MODIS and Saildrone data which refer to a specific acquisition time and day, respectively. As a consequence, the three datasets also have a temporal mismatch with respect to the SST values they measure: MUR includes only nighttime measurements, and MODIS level-2P SST corresponds to a specific time which we tried to be as close as possible to a nighttime measurement to coincide with the MUR SST. However, it was not always possible to find good MODIS data due to the high cloud contamination problems during this period of the year. The Saildrone data corresponded to a whole 24 h time window, including both day and night measurements, with the assumption that SST variations measured by the sensor within a day are negligible.

### 4.5. Assessment of the MODIS Quality Levels and the MUR Interpolation

Regarding the assessment of the characterization of the quality levels of the MODIS products, the characterization of the bad-quality pixels is generally accurate. As can be seen from the analysis of the SST and the gradients, pixels characterized as bad-quality pixels during the production of the level-2 MODIS product are indeed bad pixels/values and are correctly excluded from the production of the gridded SST maps. These areas consisting mainly of bad pixels are those close to the coast, and therefore we can conclude that the characterization of the MODIS quality level seems to be valid in areas close to the coast that

are covered by clouds. This result confirms the low quality of MODIS level-2 data that we can obtain in coastal areas due to factors such as cloud cover and erroneous atmospheric correction [35].

The results for the medium and good-quality trajectories imply that some pixels have been misclassified as medium or good-quality pixels when they appear to correspond to bad-quality pixels. Conversely, some pixels have been classified as bad-quality when they appear to correspond to medium or good-quality pixels. As a result, they are erroneously included or excluded in the production of the gridded SST maps, respectively, during the processing of the MODIS level-2 products. Furthermore, the misclassification of some pixels as "bad-quality pixels" results in data loss and interpolation for the creation of the SST maps in pixel areas that is not needed. Therefore, the interpolation to produce the level-4 MUR product contains errors which can explain the increased RMSE and MAE values for the MUR in some good and medium-quality trajectories. This inaccurate interpolation in some pixels indicates that the level-2 MODIS product may be more accurate than the corresponding level-4 MUR, even though the former is a level-2 dataset in a primary data processing step with some sources of error not yet removed. In addition, the process of assigning quality levels to each pixel is an area of ongoing research.

An interesting potential study which was not investigated in this study is the assessment of the quality levels relevant to the distance from the coast.

### 4.6. Comparing Both Gradient Approaches

The use of a larger pixel window around the Saildrone trajectory within the kriging-based approach was made to avoid using only the pixels within the Saildrone trajectory and to have a better representation of the area around the trajectory. Using only the pixels within the trajectory and considering that the Saildrone trajectories may contain shifts due to the wind (see Section 4.3), inaccuracies in the x and y coordinates can lead to incorrect estimates of gradients using Formula (1). In fact, the results of the kriging-based approach are more promising, as they generally show lower RMSE values for the MODIS products than the results of the pixel-based approach.

In addition, the kriging interpolation method is useful because the Saildrones are mainly launched near the coast and therefore there is a lack of data far away from the coast. By interpolating the SST sampled Saildrone values in areas away from the coast, we "generate" in situ data in areas where we do not yet have Saildrones deployments. This allows us to produce new datasets for the validation of satellite products, with the high spatial and temporal resolution of Saildrones. Our work can also indicate to future researchers where are the areas that need to be validated in more detail and therefore where in situ instruments need to be deployed. Such areas appear to be those far away from the coast. For the areas close to the coast, we have shown that the quality of level-2 MODIS is very low (very high or complete cover from bad pixels) and that the characterization of the quality levels of MODIS is already generally accurate.

More advanced interpolation approaches, such as taking into account correlations between Saildrone and satellite products could be further explored in future work.

The methods used in this study can also be applied to other study areas as well. Our study provides a basis for investigating the magnitude of phenomena such as coastal upwelling, especially with the integration of additional parameters such as chlorophyll-a provided by satellites and Saildrones. Additionally, our study was focused on the summer months, so it could be interesting to replicate our work during the winter months when the cloud cover is potentially lower and we, therefore, expect level-2 MODIS SST data of better quality.

## 5. Conclusions

From the comparisons between Saildrone and satellite products using the pixel-based SST approach, from all the trajectories we derived a mean RMSE of 0.86 and 4.64 K for MUR and MODIS respectively (excluding the outlier). From the gradient approaches, we derived a mean RMSE of 0.19 and 0.84 K/km for MUR and MODIS for the pixel-based approach, and a mean RMSE of 0.23 and 0.71 K/km for MUR and MODIS for the kriging-based approach. These results for MUR are promising considering the results of previous studies and the influence of the dynamics of the coastal upwelling over the California Coast. The results for MODIS show rather high inaccuracies compared to the corresponding MUR products and other studies over different study areas. The high inaccuracies for MODIS deriving from our study highlight the poor-quality of MODIS level-2 data near the coast, especially during summer.

Our comparisons with Saildrone data also showed that for the bad-quality trajectories the MUR interpolated product was more accurate than the MODIS. For the medium and good-quality trajectories, MUR was generally more accurate than MODIS, with some exceptions. These exceptions reveal possible inaccuracies within the processes of characterizing the quality level of MODIS pixels and the interpolation process for producing the MUR level-4 gridded product.

The pixel-based gradient approach gave similar results to the kriging-based approach, with the kriging-based approach being more accurate in terms of RMSE for MODIS. Kriging interpolation is an approach to generate SST data in areas where Saildrone deployments are still lacking, and to indicate where future campaigns should be conducted. The latter is important as more data acquisitions away from the coast are needed to better validate the satellite products.

The characterization of MODIS quality levels appears to be generally valid for the bad-quality pixels especially in coastal areas, but less valid for the medium and bad-quality pixels. Further research on the assessment of the MODIS quality levels is needed, for example in relation to distance from the coast.

Considering all the above-mentioned results, the Saildrone technology proved to be efficient in validating ocean satellite products due to its high spatial and temporal sampling capabilities compared to traditional in situ measurement techniques. This current study shows that high spatiotemporal in situ data can be successfully used to validate ocean SST products, both for pixel-level SST values and SST gradients. There are promising areas for further research, such as the use of Saildrone data to detect sub-pixel SST variations and to downscale satellite products using machine learning techniques.

**Author Contributions:** K.K.: Methodology; Software; Validation; Investigation; Data curation; Writing—Original draft. P.B.: Conceptualization; Writing—review and editing; Supervision; Project administration; Funding acquisition. J.V.-C.: Conceptualization; Methodology; Writing—review and editing; Supervision; Project administration. All authors have read and agreed to the published version of the manuscript.

**Funding:** The project was supported through a Fonds de donations grant from the University of Neuchatel. Kalliopi Koutantou and Philip Brunner were supported by the University of Neuchatel and Jorge Vazquez-Cuervo from NASA Jet Propulsion Laboratory, California Institute of Technology.

**Data Availability Statement:** All the satellite data is accessible through NASA's Physical Oceanography Distributed Active Archive Center (PO.DAAC). The Saildrone data can be accessed here: https://data.saildrone.com (accessed on 1 June 2022).

**Acknowledgments:** The research was carried out at the Jet Propulsion Laboratory, California Institute of Technology, under a contract with the National Aeronautics and Space Administration (80NM0018D0004). The authors would like to thank Philippe Renard and Julien Straubhaar for their assistance in the geostatistical simulations.

**Conflicts of Interest:** The authors declare no conflict of interest.

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
