# Peer review of "Validation of NASA Sea Surface Temperature Satellite Products Using Saildrone Data"

_remotesensing, doi:10.3390/rs15092277_

Round 1

Reviewer 1 Report

This is a useful quantification of high-resolution satellite SST uncertainties in a near-coastal region, based on comparisons to in-situ data from saildrones.  It will be useful for users of the satellite data for research and other applications. However, I have some questions about the methods and interpretation of the results that need to be addressed.

The co-location of satellite and saildrone data is described, but how far apart in time are they allowed to be? All that is mentioned is that the periods of June 27-30, July 7, and August 11 are used. Are all data averaged during those periods, during one-day periods, or periods based on when saildrone data are available in the grid boxes? What are the temporal distributions of saildrone and satellite data during the study periods? Is aliasing of the diurnal cycle not an issue?

How meaningful is it to compare the satellite data with low-medium quality to the saildrone data? Isn't it expected that the agreement will be worse? Does anyone use the satellite SST data when the quality level is low? You found that sometimes the RMSE and MAE are larger for medium-quality than bad-quality. However, this could be due to multiple factors, including: misclassification of quality levels in MUR (though I'm not an expert on the data, so I don't know how/why this would happen if the quality levels are assigned objectively), and spatiotemporal differences in MODIS and saildrone SST measurements (even if the quality level is higher, if there's a strong gradient or temporal variability and the MODIS and saildrone measurements are farther apart, the RMSE/MAE can be larger). An objective measure of the representativeness of the saildrone data for the pixel-averaged MODIS data would help interpretation. For example, if there are more saildrone measurements across a wider range of a MODIS pixel, I would expect the agreement to be better. Have you considered anything like this? At least there should be more explanation of how/why MUR would use MODIS data with bad-quality flags. That is puzzling to me as a non-expert.

There are numerous places in the manuscript where the figure/reference text did not display properly. Those need to be fixed. 

There are some minor edits needed, so I recommend a thorough proof-reading by the authors. For example, on lines 33-35 (sentences separated by comma instead of period):

"However, the coarse spatiotemporal resolutions of the available satellite data might not be adequate for small-scale assessments of ocean phenomena, Moreover, coastal areas are often covered by clouds thus full spatial coverage is not available."

Reviewer 2 Report

My comments are in the PDF attachment uploded to the Journal's Web site.

Reviewer 3 Report

The MS is dedicated to the independent evaluation of the accuracy of two SST remote sensing data sets -  MODIS level-2 and Multi-scale Ultra-high Resolution (MUR) level-4 products, using the novel in-situ data sets from Saildrone. The data are compared pixel-wise; prior to that interpolation is made for irregular in-situ data and MODIS Level 2 data. Sophisticated data quality flagging is included in the comparison procedures. Such research is interesting and of high importance.

The manuscript is well written and contains systematically presented results. In particular, the MS has a good introduction that can be used also for educational purposes. The main body of the MS needs some improvement before publication.

The main problem is with outliers in the processed MODIS data. Although one outlier in MODIS statistics has been removed, there are still a number of unrealistically high RMSE and MAE values for MODIS. Can these MODIS values be identified and removed prior to the comparison with Saildrone data? I mean temporal and spatial changes in MODIS temperature should not exceed some known values; if they exceed then the values should be flagged as bad. I cannot find an oceanographic reason why the RMSE and MAE of two SST determinations in a close time and space window would exceed 5 deg C in only one data set. If there are upwelling filaments, they should be present on the both data sets, if MODIS level 2 processing is adequate. Susceptible data clusters can be found on trajectories 3, 7, 14, 15, 18, 21, 22, 23 as seen in Figs. 3 and 6. Inclusion of bad data causes very high RMSE and MAE of MODIS data compared to the MUR data. This ambiguity has to be explained, or better, resolved.

There are some minor remarks as well.

1) The presentation is a bit biased towards the US conditions. The term “West Coast” has many meanings. Just one example is used in Australia, see “West Coast Council” https://www.westcoast.tas.gov.au/.

2) The units without space like 0.5m have to be written with a space 0.5 m.

3) Fig. 1 composition reminds the snapshot from the slide of the conference presentation. It could be converted into the scientific paper style.

4) Lines 251-273 describe using the Kriging interpolation method. Why was this method selected? Since Kriging is equivalent to optimal interpolation, it should be interesting to know about the used statistical properties of physical origin, namely noise-to-signal ratio, horizontal correlation scale, and shape of the correlation function (including possible anisotropy). Although Kriging is widely used in the remote sensing of rather static terrestrial features, applications of highly dynamic ocean conditions tend to prefer the optimal interpolation formulation.

5) Lines 307-312 describe selection of “good-quality”, medium-quality” and ”bad-quality” trajectories. It should include reasoning why the criteria 30% was selected, and whether some experiments were made to justify this choice.

6) Statistical properties of MUR and MODIS, as given in the Figure legends, are nearly indistinguishable in Figs. 3 and 6. The value points of these two data sets should have clearly different labels.

7) Temperature and its statistical properties are mainly written in Kelvin units, but sometimes Celsius occurs as well. There should be one unit through the MS.

Round 2

Reviewer 2 Report

(1) The screenshots are the same. The reviewer does not condone the authors' usage of a massive amount of screenshots in lieu of publication-quality figures. 

(2) Replace Kelvin with Celsius degrees everywhere in the text and in the figures. Use the degree sign instead of "degree."

I am not going to re-review the revised manuscript until the above changes are made.
